# Non-Invasive Diagnostic of NAFLD in Type 2 Diabetes Mellitus and Risk Stratification: Strengths and Limitations

**DOI:** 10.3390/life13122262

**Published:** 2023-11-27

**Authors:** Alina Boeriu, Daniela Dobru, Crina Fofiu

**Affiliations:** 1Gastroenterology Department, University of Medicine Pharmacy, Sciences, and Technology “George Emil Palade” Targu Mures, 540142 Targu Mures, Romania; aboeriu@gmail.com; 2Gastroenterology Department, Mures County Clinical Hospital, 540103 Targu Mures, Romania; 3Internal Medicine Department, Bistrita County Clinical Hospital, 420094 Bistrita, Romania

**Keywords:** diabetes, steatosis, fibrosis, grading, monitoring, biomarkers, imaging

## Abstract

The progressive potential of liver damage in type 2 diabetes mellitus (T2DM) towards advanced fibrosis, end-stage liver disease, and hepatocarcinoma has led to increased concern for quantifying liver injury and individual risk assessment. The combination of blood-based markers and imaging techniques is recommended for the initial evaluation in NAFLD and for regular monitoring to evaluate disease progression. Continued development of ultrasonographic and magnetic resonance imaging methods for accurate quantification of liver steatosis and fibrosis, as well as promising tools for the detection of high-risk NASH, have been noted. In this review, we aim to summarize available evidence regarding the usefulness of non-invasive methods for the assessment of NAFLD in T2DM. We focus on the power and limitations of various methods for diagnosis, risk stratification, and patient monitoring that support their implementation in clinical setting or in research field.

## 1. Introduction

Liver damage secondary to metabolic disorders represents an increasing pathology worldwide. Considering its progressive nature, from non-alcoholic fatty liver (NAFL) towards non-alcoholic steatohepatitis (NASH), liver fibrosis, cirrhosis, or hepatocellular carcinoma (HCC), early diagnosis, therapeutic interventions, risk stratification for disease progression, and patient monitoring are of paramount importance.

Data from epidemiologic studies show that approximately 20% of non-alcoholic fatty liver disease (NAFLD) patients will develop NASH (liver inflammation and hepatocyte damage), which carries the potential to progress to advanced fibrosis (20% among NASH cases) and even hepatocellular carcinoma. Estes et al. estimated a further increase of up to 29% of NASH cases at risk of developing advanced fibrosis and cirrhosis by 2030, against the backdrop of the rising prevalence of diabetes mellitus (DM) in the United States [1].

Published evidence suggests a close relationship between NAFLD and T2DM. An increased rate of fatty liver has been detected among diabetic patients (>60%), even in those with normal aminotransferase levels [2,3]. This highlights the importance of early screening and implementation of prevention measures to control disease progression. A high prevalence of fibrosis has been reported in patients with T2DM [4], and epidemiological studies have indicated their increased risk of developing end-stage liver disease and HCC [5,6,7]. The coexistence of NAFLD and diabetes elevates the risk of chronic complications, such as cardiovascular disorders (CVD), chronic kidney disease (CKD), diabetic microvascular complications (nephropathy, retinopathy), and both sensorimotor and autonomic neuropathy [8,9,10,11,12]. Moreover, diabetes increases the risk of NAFLD progression to high-grade fibrosis and cirrhosis [13].

In addition to detecting and accurately quantifying the degree of steatosis, assessments of inflammation, hepatocyte injury (ballooning), and fibrosis in NAFLD are crucial to evaluate the patient’s risk for severe disease and adverse outcomes. Recent data from a large Swedish population-based cohort of 10,568 adults showed that biopsy-confirmed NAFLD, ranging from simple steatosis to NASH, non-cirrhotic fibrosis, and liver cirrhosis, was significantly associated with increased overall mortality, especially from extra-hepatic cancers and cirrhosis. The risk of death progressively increased with worsening liver histology [14]. The severity of fibrosis is a strong predictor of disease-specific mortality in NAFLD, with the highest mortality rate observed among patients with fibrosis stages 3 or 4 [15].

After the initial assessment of the severity of histological lesions, regular monitoring is required in high-risk patients to evaluate either disease progression or improvement of histological parameters following therapeutic and lifestyle interventions. Although liver biopsy remains the gold standard for NAFLD assessment, its widespread use for diagnosis and monitoring is limited due to its invasive nature, which implies patient discomfort, procedural risks, sampling errors, and interobserver variability in biopsy interpretation [16,17,18,19,20].

Over the years, non-invasive diagnostic tests, including serum biomarkers, predictive scoring systems, and imaging-based techniques, have emerged as practical alternatives to histologic analysis. The accuracy of various non-invasive methods in evaluating liver changes in NAFLD, selecting high-risk patients, and testing the efficacy of different therapeutic regimens has been assessed against liver biopsy as the reference standard. The NAFLD activity score (NAS) is a histological scoring system that sums up the scores of steatosis, lobular inflammation, and hepatocellular ballooning. Developed by the NASH Clinical Research Network, it is currently employed in research trials. A NAS ≥ 4 and fibrosis stage ≥ 2 (clinically significant fibrosis) are indicative of progressive NASH, which carries an elevated risk for end-stage liver disease and mortality [21,22].

Given the rising burden of the disease in diabetic patients, various invasive and non-invasive methods and strategies for the diagnosis and follow-up of NAFLD have been developed. This review focuses on the non-invasive assessment of NAFLD in T2DM, highlighting the strengths and limitations of various diagnostic methods that support their implementation in current clinical practice or research trials.

## 2. Serum Biomarkers and Scoring Systems

### 2.1. Blood-Based Tests for Diagnosis of Simple Steatosis

Liver enzymes, such as alanine aminotransferase (ALT) and aspartate aminotransferase (AST), may be elevated in fatty liver disease. However, most NAFLD cases show values within normal ranges [23,24]. In a multiethnic, population-based study including 2287 subjects, Browning et al. demonstrated that 79% of individuals with hepatic steatosis had normal levels of serum alanine aminotransferase [25]. An Italian study that reviewed the histological data from liver biopsies of 458 patients with NAFLD showed that NASH was diagnosed in 59% of patients with normal ALT. Therefore, the study confirmed that NAFLD patients with normal ALT are at risk of progressive hepatic disease, and normal ALT cannot be considered a valuable criterion to eliminate the need for liver biopsy [26].

Several population-based studies have described elevations in liver enzymes, including gamma-glutamyltransferase (GGT) in T2DM patients. However, a precise causal association between NAFLD and the aforementioned abnormal tests could not be demonstrated. Consequently, liver enzymes alone are not reliable predictors of fatty liver [27,28,29,30].

The NAFLD Liver Fat Score (NLFS), calculated based on metabolic syndrome, T2DM, fasting serum insulin, and fasting serum AST/ALT ratio (AAR), evaluates hepatic fat content. It has shown good accuracy in diagnosing NAFLD. Kotronen et al. demonstrated an 86% sensitivity and 71% specificity in predicting increased liver fat content, using a cut-off of −0.640 [31]. A more recent analysis including obese patients showed even higher sensitivity and specificity of 95%. However, different stages of NAFLD cannot be distinguished, and the need for serum insulin might limit the wider use of NLFS in daily practice [32].

The Fatty Liver Index (FLI) is a simple test combining BMI, waist circumference, serum triglycerides, and GGT. Cut-off values ≥ 60 rule in fatty liver, with 86% sensitivity and 87% specificity (area under the receiver operating characteristic [ROC] curve [AUC] = 0.85) [33]. Recently, a Korean population-based, cross-sectional study demonstrated the role of FLI as a screening tool to detect the presence of metabolic syndrome (MetS) in NAFLD patients, at a cut-off of 20 (AUC 0.849, sensitivity 0.828, and negative predictive value NPV 91.9%) [34].

The Hepatic Steatosis Index (HSI) is a simple formula derived from a logistic regression model, which rules in NAFLD at a cut-off value > 36.0 with a specificity of 92.4% (area under the receiver operating characteristic curve AUROC of 0.812). It includes gender, history of T2DM, body mass index (BMI), ALT, and AST. The test plays a significant role in selecting patients for ultrasonography [35]. Fennoun et al. specifically examined the role of HSI in screening for NAFLD in T2DM. Their results confirmed the good accuracy of the HSI, with a sensitivity of 89.55% and specificity of 95.24% (AUROC of 0.979) [36].

Similar to FLI, the Lipid Accumulation Product (LAP) is another biomarker score with high diagnostic accuracy for identifying NAFLD in the general population (AUC of 0.843 in men and 0.887 in women). LAP was first studied in diabetes patients and was calculated based on waist circumference and fasting plasma triglyceride levels. In a large cross-sectional study, Dai et al. demonstrated a sensitivity and specificity of 77% and 75% in men for a cut-off of 30.5, respectively, and 82% and 79% in women for a cut-off of 23.0, respectively [37]. Although studies have shown that LAP could help in selecting subjects for liver ultrasonography, it still requires further validation before being used in a clinical setting [38,39].

SteatoTest (ST) represents a panel of more specialized parameters, comprising serum bilirubin, GGT, alpha-macroglobulin, haptoglobin, ALT, apolipoprotein A1, BMI, total cholesterol, triglycerides, and glucose, adjusted for age and gender. It has good accuracy in predicting hepatic steatosis, but being a commercial panel, it cannot be easily used in current practice [40].

Overall, the use of these tests has not been demonstrated to provide additional information beyond that obtained from routine laboratory tests and ultrasonographic evaluation in diabetic patients.

### 2.2. Blood-Based Tests for Diagnosis of NASH and Fibrosis

Given that referring all diabetic patients to liver specialists is not feasible in real-life settings, selecting NAFLD individuals at a higher risk for disease progression and significant fibrosis is essential [41].

Although a correlation between aminotransferase levels and the degree of hepatic fibrosis detected on biopsy could not be demonstrated, Verma et al. showed that an increased ALT value > 2 times the upper normal limit has a 50% sensitivity and 61% specificity for NASH detection. Conversely, ALT values tend to decrease as fibrosis progresses to advanced fibrosis/cirrhosis, leading to an increased AST/ALT ratio. Therefore, recent studies have included this ratio in diagnostic scores of advanced fibrosis [42].

Serum cytokeratin (CK)-18, a marker of hepatocyte apoptosis, is the most studied single parameter to predict NASH in NAFLD patients [43]. However, when studying its clinical value in a multiethnic NAFLD population, Cusi et al. highlighted its limited sensitivity and the absence of a clear cut-off value, making it inadequate as a screening tool [44,45]. CK-18 was also included in the NASH Diagnostic Panel, alongside adiponectin and resistin, but it was not found to be effective in predicting fibrosis [46].

The AST/platelet ratio index (APRI), primarily studied in hepatitis C patients, demonstrated good accuracy in predicting hepatic fibrosis in viral hepatitis compared to liver biopsy, with cut-off values of 0.5 for fibrosis and 1.5 for cirrhosis [47]. However, its diagnostic performance for significant and advanced fibrosis in NAFLD is reduced (AUROC of 0.564 and 0.568, respectively) [48].

FIB-4 is a simple score that includes age, AST, ALT, and platelet count. The test was developed to detect patients at low risk for advanced fibrosis, with a 90% NPV and 70% sensitivity to exclude advanced fibrosis at a cutoff < 1.45 [49]. Repeated FIB-4 measurements can be used in the clinical setting to detect and monitor patients with an increased risk of developing advanced liver disease, as reported by Hagström et al. Cut-off values used to discriminate risk groups were: <1.30 for low risk, 1.30–2.6 for intermediate risk, and >2.67 for high-risk patients [50].

The NAFLD fibrosis score (NFS), which includes age, hyperglycemia, BMI, platelet count, albumin, and AST/ALT ratio, has demonstrated good diagnostic performance for advanced fibrosis using rule-in and rule-out cutoff values (0.676 and −1.455, respectively). Angulo et al. validated this scoring system in a multicenter study focused on liver biopsy-confirmed NAFLD. They described an accurate lower cut-off in ruling out advanced fibrosis (NPV of 93% and 88% in the estimation and validation groups, respectively), sparing 75% of patients from liver biopsy [51]. In a large sample size study conducted by Ciardullo et al., NFS, along with FIB-4 and AST/ALT ratio, showed a good correlation with the prevalence of CVD and CKD in T2DM patients [41].

Laboratory fibrosis tests demonstrated good diagnostic performance for advanced fibrosis and for assessing progression to advanced fibrosis in NAFLD (cross-validated C-statistic of 0.82, 0.81, and 0.80, respectively, for APRI, FIB-4, and NFS). At 90% sensitivity, the NPV for detecting advanced fibrosis was 93%, 91%, and 91% for FIB-4, APRI, and NFS, respectively, highlighting their utility in ruling out advanced fibrosis [52]. However, published evidence suggests the lower performance of NFS in diabetic patients, where FIB-4 might be a better choice [53,54].

The Enhanced Liver Fibrosis (ELF) test, which combines age and circulating extracellular matrix components such as hyaluronic acid (HA), amino-terminal propeptide of type III procollagen (PIIINP), and tissue inhibitor of metalloproteinase-1 (TIMP-1), was successfully used to identify severe fibrosis in NAFLD (sensitivity of 86.7%, specificity of 92.5%, positive predictive value PPV of 72%, and negative predictive value NPV of 97% for a cut-off value ≥ 9.8) [55]. Various thresholds have been used to select patients requiring close monitoring for disease progression. An ELF score ≥ 11.3 can be used as a predictor of future liver-related events [56].

FibroTest, mostly validated in chronic hepatitis C, includes serum α2-macroglobulin, apo A1, haptoglobin, total bilirubin, and GGT. Ratziu et al. tested the diagnostic value of this scoring test for predicting advanced liver fibrosis in NAFLD patients. They reported a 90% NPV at a cut-off of 0.30 (77% sensitivity) and a 73% PPV at a cut-off of 0.70 (98% specificity) [57].

By assessing Fibromax (Steato-Acti-Nash-FibroTest) in T2DM patients, Bril et al. demonstrated that some of these panels underperform in this condition. Therefore, results from studies conducted in non-diabetic populations should not be extrapolated to patients with T2DM, who may require different predictive models [58].

The same working group that developed FibroTest, ActiTest, and SteatoTest (Biopredictive, Paris, France), which proved to have high predictive values for diagnosing fibrosis, activity, and steatosis, respectively, searched for a simple biomarker that could predict NASH in NAFLD patients. As a result, NashTest, which includes the FibroTest components plus AST, cholesterol, triglycerides, glucose, and BMI, was validated for detecting NASH in NAFLD, with a specificity of 94% (66% PPV) and a sensitivity of 33% (81% NPV) [40,57,59].

### 2.3. Emerging Serum Non-Invasive Biomarkers

New diagnostic non-invasive tools, such as lipidomic, metabolomic, and proteomic biomarkers, have been developed to diagnose patients at higher risk for severe disease and liver-related complications.

#### 2.3.1. Proteomic Analysis

By analyzing a group of liver biopsy specimens from bariatric surgery patients, Younossi et al. described different expressions of several genes in liver tissue and serum protein peaks in NAFLD patients. The authors demonstrated an overall downregulation of phase II detoxification enzymes (Mu-class glutathione S-transferases and cytosolic sulfotransferase isoform 1A2 (SULT1A2)) and upregulation of cell survival and liver regeneration genes in the early stages of NASH. Increased expression of genes related to the activation of stellate cells, fibrogenesis, and detoxification pathways was observed in late-stage NASH. Furthermore, they described 12 serum protein peaks in NAFLD patients, which were differently expressed depending on the severity of NAFLD/NASH [60].

Using the same mass spectrometry method improved with a label-free quantitative proteomics (LFQP) approach, Bell et al. identified over 1700 serum proteins. Of these, 605 showed significant changes in NAFLD compared to the control group. Moreover, 55 of those 605 proteins had different expressions between simple steatosis and the NASH F3/F4 group, and 15 between NASH and NASH F3/F4, respectively [61].

#### 2.3.2. Metabolomics

Recently, several studies have focused on analyzing changes in the plasma metabolome in subjects with fatty liver. Using mass spectrometry, Kalhan et al. showed significant changes in bile acids, glutathione metabolism, lipid, and amino acid metabolism, changes that were more pronounced in NASH than in simple steatosis [62]. Of these, pyroglutamate was found to be the most promising marker in distinguishing NASH from simple steatosis, with a sensitivity and specificity for NASH diagnosis of 72% and 85%, respectively [63].

#### 2.3.3. Lipidomics

Lipidomics represents a promising diagnostic strategy in fatty liver, proving a high accuracy in discriminating between NAFLD and normal liver (0.94 sensitivity and 0.57 specificity, respectively) and also between NAFLD and NASH (0.70 sensitivity and 0.81 specificity, respectively) [64]. Despite these encouraging results, the use of omics in daily clinical practice is limited due to their complexity and the required laboratory expertise, making them more suitable for research settings.

## 3. Imaging Techniques

Imaging methods have evolved over time to quantify liver steatosis and fibrosis, aiming to estimate the severity of liver changes throughout the organ (Table 1). This evolution seeks to overcome the limitations arising from the histological analysis of a small tissue biopsy sample.

### 3.1. Ultrasonography

#### 3.1.1. Qualitative Assessment

Abdominal ultrasonography (US) is an effective method for screening and monitoring liver changes in patients with T2DM. It offers the advantages of lower cost and increased availability compared to other imaging methods. The primary features of liver steatosis include increased liver echogenicity (bright liver) and beam attenuation, which hinder the proper visualization of intrahepatic vessels, bile ducts, the deeper part of the liver, and the diaphragm. Grade 0 indicates the absence of steatosis, increased liver echogenicity more than the renal cortex corresponds to grade I steatosis, liver echogenicity that obscures portal venous walls corresponds to grade II steatosis, and increased liver echogenicity with poor visualization of portal venous walls, diaphragm, and posterior parts of the right lobe corresponds to grade III steatosis [65].

Numerous studies have assessed the diagnostic accuracy of US and its correlation with histological findings, revealing suboptimal performance in detecting mild steatosis (Table 2).

Abdominal ultrasound demonstrated 100% sensitivity in diagnosing moderate and severe hepatic steatosis (>33% fat on liver biopsy) [66], but its diagnostic accuracy for steatosis of any grade was lower (65% sensitivity, 77% specificity) [67]. In a meta-analysis that included 4720 participants, Hernaez et al. reported an 84.8% sensitivity, 93.6% specificity, 13.3 positive likelihood ratio, and 0.16 negative likelihood ratio of US for diagnosing moderate to severe liver steatosis, using histology as the reference standard [68].

The performance of US has been compared to other imaging methods to determine the ideal diagnostic strategy in NAFLD (Table 2). A meta-analysis by Bohte et al. reported limited diagnostic accuracy of US for evaluating hepatic steatosis (73.3–90.5% sensitivity, 69.6–85.2% specificity). Liver fat is better quantified by magnetic resonance imaging (MRI) and magnetic resonance spectroscopy (MRS) than by US or computed tomography (CT), especially in cases of mild steatosis (<30% fat in the liver). However, the heterogeneity of the reports included in this analysis might influence the accuracy of the final data [69]. Palmentieri et al. reported that the bright liver echo pattern is an accurate indicator of steatosis > 30% (with 91% sensitivity and 93% specificity), either alone or in association with posterior attenuation and/or focal fatty sparing areas (with 89.7% sensitivity and 100% specificity) [85].

A similar appearance of increased liver echogenicity can be detected in other liver disorders, including fibrosis. Beam attenuation due to fat from subcutaneous tissue in obese patients may overestimate the steatosis grade [86]. Interobserver and intraobserver variability, as well as the lack of correspondence with histologic severity, are current limitations of US for accurately estimating liver damage (steatosis, inflammation, and fibrosis) and assessing the risk for disease progression [66,87]. Nonetheless, an initial, easy-to-perform evaluation of patients with T2DM using US is a common practice in clinical settings, as recommended by current international guidelines [88].

#### 3.1.2. Semi-Quantitative US Methods

Conventional US provides only a qualitative assessment of liver steatosis and is both operator and machine-dependent. New computerized quantitative methods to evaluate liver steatosis have been developed to overcome operator subjectivity, with promising results (Table 2). Specialized software can calculate the hepatorenal index (HRI), representing the ratio between the average liver brightness and the average kidney brightness. An HRI of 1.34 or higher showed 92% sensitivity and 85% specificity for detecting liver steatosis > 5% [70]. Recently, Johnson et al. demonstrated the usefulness of HRI (>95% PPV) in estimating steatosis of various degrees (steatosis < 5% and steatosis > 10%, respectively) [89]. The presence of concomitant fibrosis limits the diagnostic performance of HRI. Stahlschmidt et al. reported a limited ability of HRI to differentiate mild from moderate-severe steatosis (fat fraction > 15% on MRS) in patients diagnosed with advanced fibrosis at US elastography (shear wave speed ≥ 1.78 m/s) [90].

Other semi-quantitative methods for liver steatosis include the Hamaguchi score and the US fatty liver index (US-FLI). The Hamaguchi score incorporates US features corresponding to steatosis: hepatorenal contrast, bright liver, deep attenuation, and vessel blurring. A score ≥ 2 indicates liver steatosis, while a score ≥ 4.0 reflects moderate-severe steatosis [91,92]. Ibacahe et al. reported the good diagnostic performance of the Hamaguchi US score ≥ 4.0 for NAFLD (AUROC of 86%) and its agreement with MRS (Kappa of 0.63). The sensitivity and specificity of the US score ≥ 4.0 for NAFLD diagnosis were 78% and 85%, respectively [71].

US methods for liver steatosis quantification have seen rapid development and expanded application in clinical practice. Emerging data show a good diagnostic performance of semi-quantitative US methods (82.2% and 100% sensitivity; 86.9% and 94.8% specificity for Hamaguchi’s score and HRI, respectively), using US quantitative fat assessment (controlled attenuation parameter CAP) as a reference [93]. An increased level of concordance between observers was reported [93]. However, all of these promising data emerged from studies with small sample sizes and require validation through larger trials.

#### 3.1.3. Quantitative US Methods

Given the limitations of conventional US, US-based elastographic methods have been developed to quantify liver steatosis and fibrosis. Liver biopsy and advanced MRI techniques have been used as reference standards to evaluate the diagnostic performance of various US methods (Table 2), in line with the World Federation for Ultrasound in Medicine and Biology (WFUMB) recommendations [94].

Shear wave elastography (SWE) methods, which include transient elastography (TE), point shear wave elastography (pSWE) using Acoustic Radiation Force Impulse (ARFI) technology, and multidimensional shear wave elastography (2D-SWE and 3D-SWE), measure the internal tissue shear deformations generated by an applied force (either a mechanical thump on the tissue surface or an ultrasound-induced focused radiation force at a controlled depth) [95]. Vibration-controlled transient elastography (VCTE, FibroScan from EchoSens, Paris, France) is a point-of-care technique that quantifies liver fibrosis by measuring the speed of shear waves generated by an external mechanical impulse. Shear wave propagation velocity through the liver increases with increasing liver stiffness, as seen in fibrosis. However, factors like postprandial status, acute hepatitis, flare of transaminases, congestive heart failure, and extrahepatic cholestasis can influence liver stiffness and the accuracy of measurements [96]. Reliable criteria for liver stiffness measurement consist of 10 valid measurements with an interquartile range/median ratio (IQR/M) < 0.30 [97].

The controlled attenuation parameter (CAP) was added to the FibroScan system to quantify liver fat content by measuring the attenuation of US waves that pass through the liver. Depending on the skin-to-liver capsule distance, either the M probe (for normal weight patients) or the XL probe (for overweight/obese patients) can be used. There are ongoing debates regarding the optimal CAP thresholds for grading hepatic steatosis, given the diversity of results from heterogeneous studies that included patients with different etiologies of liver disease. Many studies have used liver biopsy as the reference standard to assess the diagnostic performance of CAP. Although CAP correlates with the histologic degree of steatosis, factors have been identified that may influence CAP values. In the meta-analysis conducted by Karlas et al., the optimal cut-off values were 248 dB/m for S ≥ 1 (AUROC 0.823), 268 dB/m for S ≥ 2 (AUROC 0.865), and 280 dB/m for S3 (AUROC 0.882). CAP values were influenced by BMI, diabetes, and liver disease etiology (NAFLD). NAFLD and diabetic patients presented higher CAP values (by approximately 10 dB/m) compared to patients with other etiologies of liver disease for the same biopsy-proven steatosis grade [72].

Studied performed in patients with NAFLD have demonstrated the limited accuracy of CAP in grading liver steatosis. In a multi-center prospective study that included 450 patients with NAFLD, Eddowes et al. demonstrated good performance of CAP for detecting liver steatosis (AUROC of 0.87 for S ≥ S1), but suboptimal performance in diagnosing moderate and severe steatosis (AUROCs of 0.77 for S ≥ S2 and 0.70 for S = S3, respectively). The Youden’s cut-off values for S ≥ S1, S2, and S3 were 302 dB/m, 331 dB/m, and 337 dB/m, respectively [73]. Similarly, in a meta-analysis that included 1297 patients, Pu et al. reported the limited diagnostic performance of CAP for severe steatosis. The pooled sensitivity, specificity, and AUROC values were: 87%, 91%, and 0.96 for S ≥ S1, respectively; 85%, 74%, and 0.82 for S ≥ S2, respectively; 76%, 58%, and 0.70 for severe steatosis, respectively (S3). The limited diagnostic value of CAP in obese patients with a skin-to-liver capsular distance greater than 25 mm might explain the challenges in estimating severe steatosis [74].

Another more recent meta-analysis, which included 2346 patients with chronic liver disease of various etiologies, confirmed the limited value of CAP in detecting moderate and severe steatosis in NAFLD (AUROCs of 0.736 for S ≥ 2, and 0.711 for S = 3, respectively). However, the accuracy in detecting steatosis of any grade (S ≥ 1) was good (AUROC of 0.819). CAP values were independently affected by the etiology of liver disease, diabetes, BMI, AST, and gender (with increased values observed in males). Even with the appropriate use of the XL probe in obese patients, CAP’s performance in staging steatosis proved to be limited in NAFLD. Based on the available data, the utility of CAP for NAFLD assessment in populations at increased risk (such as those with type 2 DM) requires additional investigation [75].

Other studies that used magnetic resonance imaging proton density fat fraction (MRI-PDFF) as the reference standard reported a good performance of CAP in detecting hepatic steatosis. The cut-off value for detecting steatosis (MRI-PDFF ≥ 5%) was 288 dB/m (AUROC of 0.80), while the cut-off value for MRI-PDFF ≥ 10% was 306 dB/m (AUROC of 0.87) [76]. Compared to MRI-PDFF performance, CAP has limited value in diagnosing moderate–severe steatosis and in monitoring changes in the degree of steatosis during follow-up [98].

In conclusion, CAP showed limited performance for grading steatosis in NAFLD (Table 2), but the method can be successfully used as a screening tool for liver steatosis [98]. The optimal cut-off values need to be better defined, and potential confounding factors such as gender, transaminases, obesity, and diabetes [75,99] must be systematically studied. Given their lower cost, accessibility, and ease of application in routine clinical practice, US-based methods can be recommended for the serial assessment of liver steatosis [100], although their performance in NAFLD is not optimal.

Non-invasive evaluation of liver fibrosis with FibroScan has become increasingly popular in NAFLD patients. The importance of liver damage screening using both CAP and liver stiffness measurement (LSM) in patients with T2DM was highlighted by Kwok et al., who reported a high prevalence of NAFLD and advanced fibrosis in these patients, especially when diabetes was associated with obesity and dyslipidemia [101]. In a large population-based study that included individuals aged ≥45 years, significant fibrosis (LSM ≥ 8 kPa) was correlated with steatosis and diabetes mellitus [102].

Compared to other non-invasive scores, LSM proved to be a better predictor of liver fibrosis in NAFLD. Optimal cut-off values for fibrosis stages ≥1, ≥2, ≥3, and 4 reported by Kumar et al. were 6.1, 7.0, 9.0, and 11.8 kPa, respectively (AUROCs of 0.82, 0.85, 0.94, and 0.96, respectively). The negative predictive value to rule out advanced fibrosis was 95% [77]. Eddowes et al. showed an increase in the diagnostic performance of TE with increasing degrees of liver fibrosis. The Youden’s cut-off values were 8.2 kPa (AUROC of 0.77), 9.7 kPa (AUROC of 0.80), and 13.6 kPa (AUROC of 0.89) for F ≥ F2, F ≥ F3, and F = F4, respectively [73].

The influence of hepatic steatosis severity on liver stiffness measurement has been a subject of debate. Petta et al. reported that severe steatosis (>66%) could lead to an overestimation of the severity of liver fibrosis in NAFLD when the M probe was used [103]. More recently, Eddowes et al. demonstrated that neither probe type nor steatosis influences LSM [73]. Similarly, Wong et al. reported that severe steatosis did not increase LSM, and the same LSM cut-offs should be used for both the M or XL probe [104].

The performance of transient elastography in diagnosing severe liver fibrosis (F3-F4) in NAFLD has been validated in numerous studies against liver histology as the reference standard. Petta et al. used cut-off values of <7.9 KPa and ≥9.6 KPa to rule out and rule in severe fibrosis, respectively. By combining LSM and NAFLD fibrosis score (NFS), the diagnostic accuracy for severe fibrosis increased, while the need for liver biopsy decreased by 50–60% [105]. Based on available data, TE is useful in selecting patients at risk for clinically significant fibrosis who deserve evaluation and monitoring in specialized centers [78,79,106], and in ruling out liver cirrhosis in NAFLD, as endorsed by the EFSUMB (European Federation of Societies for Ultrasound in Medicine and Biology) guidelines [95].

ARFI technology (Acoustic radiation force impulse), which integrates B-mode US and elastography, has demonstrated promising results in the evaluation of liver fibrosis. In a systematic meta-analysis, Liu et al. reported an 80.3% summary sensitivity, 85.2% summary specificity, and a 30.13 pooled diagnostic odds ratio for detecting significant fibrosis in NAFLD [107]. When comparing the diagnostic performance of TE with pSWE, TE demonstrated superiority in diagnosing significant fibrosis (AUROCs for diagnosis of fibrosis stage ≥F2, ≥F3, and F4 were 0.83, 0.83, and 0.89, respectively, for TE versus 0.72, 0.69, and 0.79, respectively, for pSWE). Both elastographic methods reported excellent intra-observer and inter-observer variability [108].

While CAP cannot be used as a prognostic marker because it does not predict liver-related events [109], LSM has proven to be an indicator of adverse outcomes [110,111]. A liver stiffness value ≥ 20 kPa is associated with an increased risk for liver-related complications [111]. Petta et al. demonstrated in a retrospective analysis, including 1039 patients with NAFLD and advanced fibrosis (F3-F4), that baseline liver stiffness measurements by FibroScan and their subsequent changes during follow-up can be used to estimate the risk for hepatic decompensation, HCC, and death. However, the lack of data regarding changes in ALT and BMI during the follow-up, as well as the absence of a standardized protocol for LSM follow-up, may be potential sources of bias [112].

Moreover, TE is a useful method for the early identification of patients with chronic liver disease at risk of developing clinically significant portal hypertension. According to the Baveno VI consensus, an LSM ≥ 10 kPa suggests compensated advanced chronic liver disease, but additional tests are required for confirmation. An LSM > 15 kPa is highly suggestive of compensated advanced chronic liver disease, while patients with liver stiffness < 20 kPa associated with a platelet count > 150,000 can avoid endoscopy since their risk of having varices requiring treatment is very low. Annual monitoring by TE and platelet count determination is recommended for these patients to reassess the need for endoscopic evaluation [113]. In a multicenter study, Petta et al. combined LSM and platelet count and used new thresholds to rule out varices needing treatment: a platelet count > 110,000/mm^3^ and LSM < 30 kPa for the M probe, and a platelet count > 110,000/mm^3^ and LSM < 25 kPa for the XL probe, respectively. This approach can reduce the need for endoscopic screening in NAFLD cirrhosis [114].

In conclusion, the most significant contribution of US elastography methods lies in the diagnosis of steatosis and advanced fibrosis, with limited performance in staging liver steatosis (Table 1 and Table 2). Screening and initial quantification of liver steatosis and fibrosis using sonographic methods in patients with an increased risk of NAFLD (including T2DM), followed by periodic reassessment, can be a reliable alternative to more invasive or expensive methods like liver biopsy or MRI-PDFF [115]. Available data show that LSM can be used for estimating the risk of liver-related complications and varices needing treatment, while the potential value in monitoring liver changes after lifestyle and medical intervention requires further research.

### 3.2. Computed Tomography

Computed tomography (CT) is a reliable method to detect liver steatosis, based on X-ray attenuation (measured in Hounsfield units, HU) due to fat accumulation in liver parenchyma. The absolute liver attenuation value, the attenuation difference between the liver and spleen, and the ratio of liver to spleen Hounsfield units (L/S ratio < 1) are CT parameters used to diagnose liver steatosis [116,117]. Although the absolute liver attenuation value correlates with the severity of liver steatosis, it may be prone to errors due to variations in attenuation values measured by CT scanners from different vendors [118]. Therefore, the difference between attenuation values of the liver and spleen has been used with good performance in diagnosing moderate to severe liver steatosis [119]. The normal value of the difference between liver and spleen attenuation (CT_L−S_) is between 1–18 HU. A CT_L−S_ value of <1 is an indicator of liver steatosis at non-enhanced CT [120].

Liver attenuation less than 48 HU has shown increased specificity (100%) for moderate to severe steatosis, with 53.8% sensitivity, 100% PPV, and 93.9% NPV [121]. The limited performance in detecting mild steatosis was reported by Bohte et al., who showed that the sensitivity and specificity of unenhanced CT for mild steatosis are lower than those for diagnosing moderate to severe steatosis (57% sensitivity and 88% specificity versus 72% sensitivity and 94.6% specificity, respectively) [69]. Both absolute and relative attenuation values (normalized with the spleen) were assessed in contrast-enhanced and unenhanced CT. Unenhanced CT proved to be a better method for predicting fat content [122], although portal-phase contrast-enhanced CT demonstrated good performance in diagnosing fatty liver [123]. However, liver attenuation can be affected by iron overload, ingestion of amiodarone, iodine contrast, glycogen overload, and hepatitis [124].

Regarding the ability to detect liver fibrosis, the CT fibrosis score, which combines the caudate-to-right lobe ratio and the diameters of liver veins, showed a significant correlation with the stage of liver fibrosis [125]. It was used with good performance for detecting both pre-cirrhotic fibrosis (83% sensitivity, 76% specificity) and cirrhosis (88% sensitivity, 82% specificity) [80]. Although CT is useful for detecting liver steatosis and advanced fibrosis, its value in the primary diagnosis of NAFLD is limited. Additionally, CT monitoring for liver changes in diabetic patients is not a common approach, given the risk of repeated exposure to ionizing radiation and limited accuracy for low-degree steatosis.

### 3.3. Magnetic Resonance Imaging

Advanced imaging methods have been proposed to improve the quantification of hepatic steatosis and for patient monitoring. Magnetic resonance imaging (MRI) techniques capture the signal intensity of water and fat protons and quantify fatty infiltration of the liver with increased accuracy compared to US and CT [126]. The methods for liver steatosis assessment are based on the chemical shift between water and fat resonance frequencies: magnetic resonance spectroscopy (MRS) and chemical shift encoded MRI (CSE MRI) [124,127]. The ratio between fat proton signals and the sum of fat and water proton signals represents the proton density fat fraction (PDFF), an imaging biomarker of tissue triglyceride concentration [128,129]. Advanced MR techniques measure the PDFF and address confounding factors to minimize errors in fat quantification: T1 bias, T2 bias, T2* decay, spectral complexity of fat, noise bias, eddy currents, J-coupling, and magnetic field strength [126].

MRS is considered the gold standard for fat quantification [129]. A high-resolution spectrum composed of water and fat resonance peaks is acquired from a single voxel sequence. However, the small sampling volume, with no possibility to generate a PDFF map, as well as the need for advanced expertise in spectra acquisition and analysis, limit the current use of MRS in a clinical setting [124,126]. On the other hand, CSE-MRI samples the entire liver parenchyma and provides a PDFF map for accurate fat quantification [130]. The heterogeneous distribution of fat in the liver and the variability of biopsy sampling prevents the accurate estimation of disease severity and adequate monitoring of disease progression. With the ability to evaluate the entire organ and make a volumetric assessment of fat content, MRI-PDFF surpasses the limitations of liver biopsy sampling. The European Association for the Study of the Liver (EASL), the European Association for the Study of Diabetes (EASD), the European Association for the Study of Obesity (EASO), and the American Association for the Study of Liver Disease (AASLD) included MRI criteria in NAFLD diagnosis. According to EASL-EASD-EASO clinical practice guidelines, >5% liver fat content on MRI-PDFF or >5.6% on MRS corresponds with NAFLD [88,131].

The strong correlation between MRI-PDFF and MRS has been demonstrated in numerous studies [132,133,134]. In a meta-analysis that included twenty-eight studies, Yokoo et al. highlighted the good diagnostic performance and reproducibility of chemical MR imaging-PDFF (with repeatability and reproducibility coefficients of 2.99% and 4.12%, respectively; strong linearity with MRS: R^2^ = 0.96) [133]. The accuracy and reproducibility of PDFF measurements across sites, vendors, and field strengths support its use in multi-center research trials [135]. PDFF showed a strong correlation with histology (r = 0.850; CI: 0.791–0.894) and triglyceride extraction (r = 0.871; CI: 0.818–0.909) in an ex-vivo validation study, with significantly less variability than histologic grading of steatosis [136].

In a meta-analysis that included six studies and 635 patients with biopsy-proven NAFLD, Gu et al. demonstrated the good capacity of MRI-PDFF to discriminate steatosis grades 0 vs. 1–3 (summary AUROC values 0.98, pooled sensitivity 0.93, pooled specificity 0.90), 0–1 vs. 2–3 (summary AUROC values 0.91, pooled sensitivity 0.94, pooled specificity 0.74), and 0–2 vs. 3 (summary AUROC values 0.90, pooled sensitivity 0.74, pooled specificity 0.87), respectively [81]. Similarly, other authors reported the high accuracy of MRI-PDFF in quantifying steatosis and in monitoring changes in the degree of steatosis over time, using histologic validation as a reference standard [82,137,138]. PDFF thresholds were 16.3% to discriminate steatosis grade 0–1 from 2–3 (83% sensitivity and 90% specificity) and 21.7% to discriminate steatosis grade 0–2 from 3 (84% sensitivity and 90% specificity) in the analysis performed by Middleton et al. [82] (Table 2).

After the initial evaluation of liver damage in diabetic patients, lifestyle changes and/or medication are prescribed in selected cases. It is important for these patients to be included in a program to monitor disease progression and therapeutic effectiveness. Since invasive procedures are not preferred by the majority of patients, a reliable non-invasive method for a proper estimation of liver changes is required. MRI techniques have proven their value in patient monitoring and have been successfully used as an objective tool to evaluate histological response after therapeutic interventions in NAFLD [130,139]. The reduction in liver fat content quantified by MRI-PDFF showed a good correlation with histological improvement in treatment trials [140,141]. Both MRS and MRI-PDFF were used to evaluate the effect of insulin glargine and liraglutide on liver fat content in T2DM patients [142].

The close relationship between diabetes and NAFLD, as well as the impact of both disorders on patient prognosis, were recently studied using MR imaging modalities for liver fat content (LFC) quantification. LFC showed a significant correlation with the severity of metabolic disturbances in patients with T2DM and was identified as a risk factor for atherosclerotic cardiovascular disease. An increase in LFC puts diabetic patients at higher risk for metabolic disturbances and chronic complications, which demonstrates the importance of accurate steatosis grading and early therapeutic interventions [143].

The complexity of imaging acquisition and data processing for PDFF measurement, the need for simplified data interpretation for clinical use, limited availability, and significant cost hinder the widespread adoption of MRI methods (Table 1). Other limitations are patient-related, such as implanted devices or claustrophobia. However, the high accuracy and reproducibility for objective assessment of liver fat make a strong case for increasing clinical and research applicability [129].

Several MRI methods have been developed for the evaluation of liver inflammation and fibrosis. Magnetic resonance elastography (MRE) offers superior diagnostic performance for diagnosing fibrosis compared to US-based fibrosis quantification (Table 2). Park et al. demonstrated the superiority of MR imaging modalities over US-based methods for steatosis and fibrosis quantification in biopsy-proven NAFLD. MR elastography was better than TE in detecting fibrosis stages 2, 3, and 4 (AUROCs of 0.89, 0.87, and 0.87 versus 0.86, 0.80, and 0.69, respectively), while MRI-PDFF outperformed CAP in diagnosing steatosis of grade 2 or 3 (AUROCs of 0.90 and 0.92 versus 0.70 and 0.73, respectively) [83]. Similarly, in a systematic review and meta-analysis, Hsu et al. showed significantly higher accuracy of MRE compared to TE in grading fibrosis: AUROCs of 0.87 vs. 0.82 for F ≥ 1 (*p* = 0.04); 0.92 vs. 0.87 for F ≥ 2 (*p* = 0.03); 0.93 vs. 0.84 for F ≥ 3 (*p* = 0.001); and 0.94 vs. 0.84 for F4 (*p* = 0.005) [84]. Optimal MRE thresholds for the diagnosis of fibrosis were 2.61 kPa for F ≥ 1, and 3.62 kPa for advanced fibrosis (F ≥ 3), respectively.

Recent studies focused on the role of MRE in risk stratification and patient monitoring. Baseline LSM proved to represent a strong predictor of cirrhosis development in NAFLD (C-statistic = 0.86). Patients with baseline LSM ranging between 4–5 kPa require close follow-up, as the risk of progression to cirrhosis for this group varies between 1.78% and 5.26% in one year. In cirrhotic patients, baseline LSM may predict future decompensation or death: the risk for poor outcomes was 9% for a 5 kPa baseline value and increased to 20% for an 8 kPa baseline value. Dynamic changes over time in LSM in NAFLD cirrhosis predicted poor prognosis: a 32% higher risk of decompensation and death for each 1 kPa increase [144]. Other data from an individual patient meta-analysis showed that the risk of liver decompensation (ascites, hepatic encephalopathy, and varices needing treatment) over three years of follow-up increased with increasing liver stiffness: patients with an LSM < 5 kPa had a 1.6% risk, those with an LSM of 5–8 kPa had a 17% risk, while those with an LSM > 8 kPa faced a 19% risk. Hepatocellular carcinoma (HCC) screening should be recommended for patients with an LSM ≥ 5 kPa because the three-year risk of incident HCC increased to >5% in this category. The MEFIB index (a combination of MRE and FIB-4) also correlated with liver-related events, HCC, and death in this analysis [145].

The major limitations of MRE are the high costs and limited availability. Acute inflammation and iron overload can affect the accuracy of liver stiffness measurement [146]. With growing interest in detecting patients at increased risk for disease progression, MR technology has evolved to address both the inflammatory and fibrotic components of NAFLD. Thus, the multiparametric magnetic resonance technique is a promising tool for assessing disease severity, risk stratification, and patient monitoring over time [147]. Iron-corrected T1 mapping (cT1) showed good performance (AUROC of 0.78) in diagnosing high-risk NASH (NAS ≥ 4 and fibrosis stage F ≥ 2) [148].

## 4. Combined Strategy for NAFLD Diagnosis, Risk Stratification, and Follow-Up in Diabetic Patients

Given the impressive data in the field of non-invasive assessment of NAFLD, a standardized strategy for diagnostic and follow-up is required. In current practice, a patient with type 2 diabetes is evaluated for the presence of liver disease by performing serologic liver tests and a conventional B-mode ultrasonography. In most situations, steatosis (NAFL) is the most common condition diagnosed by this approach. According to the 2021 update of the EASL clinical practice guidelines, conventional US remains the first-line method to assess liver steatosis in clinical settings. Serum steatosis scores (fatty liver index FLI, hepatic steatosis index HSI, the SteatoTest, and the NAFLD liver fat score NAFLD-LFS) are not currently recommended for the diagnosis of steatosis [149]. Additional work-up for the assessment of liver damage are required for risk stratification. EASL-EASD-EASO clinical practice guidelines stress the need to diagnose the progressive form of NAFLD and advanced fibrosis in type 2 diabetes mellitus [88] because the high-grade fibrosis carries an increased risk for liver-related events and death in NAFLD [150].

Laboratory-based biomarkers are useful tools for the primary assessment of fibrosis (NFS, FIB-4, ELF, or FibroTest). The FIB-4 score has proven useful in assessing the risk for advanced fibrosis in diabetic patents [52]. According to EASL and AASLD guidelines, low-risk patients (FIB-4 < 1.3) can be managed in primary care, and fibrosis scores should be done periodically (every 1–2 years) for risk reassessment [149,151]. Patients with FIB-4 ≥ 1.3 should be referred to a specialized center for secondary risk stratification using transient elastography. LSM < 8 kPa indicates a low-risk patient, while LSM ≥ 8 kPa corresponds with intermediate/high risk. Additional patented serum tests are required in this last category to confirm advanced fibrosis (F3-F4): ELF (cut-off 9.8), FibroMeter (cut-off 0.45), or Fibrotest (cut-off 0.48). In cases of discordant results or lack of serum tests, a liver biopsy is indicated for the final diagnosis [149]. The FIB-4 > 2.67 and LSM > 12 kPa indicate high-risk patients with clinically significant fibrosis, who require referral to gastroenterology/hepatology care [151,152]. In specialized centers, patients with intermediate/high risk should undergo liver biopsy or MRE for accurate fibrosis staging. Combined strategies, including serologic and imaging biomarkers, have been proposed to reduce the need for liver biopsy, although validation for various populations and settings is necessary before their large-scale implementation. Annual monitoring using non-invasive tests is recommended for patients with advanced-stage NAFLD, while those with cirrhosis should be monitored at 6-month intervals [149].

The efficacy of the sequential approach to NAFLD patients is supported by a recent meta-analysis that assessed the performance of non-invasive tests for diagnosing advanced fibrosis. Diagnostic performance was good for LSM by VCTE (AUROC of 0.85), but reduced for serologic non-invasive tests (AUROCs of 0.76, 0.73, 0.70, and 0.64 for FIB-4, NFS, APRI, and AST/ALT, respectively). The algorithm that uses the sequential combination of FIB-4 and LSM-VCTE lower cut-offs to rule out advanced fibrosis (FIB-4 < 1.3; LSM < 8.0 kPa) and upper cut-offs to rule in cirrhosis (FIB-4 ≥ 2.67; LSM ≥ 10.0 kPa) showed 66% sensitivity and 86% specificity, reducing the need for liver biopsy to 33%. By using higher upper FIB-4 cut-offs (≥3.48) and LSM cut-offs (≥20.0 kPa), the specificity for detecting cirrhosis increased to 90%, with 38% sensitivity, while the proportion of patients needing liver biopsy was reduced to 19% [153]. Because many studies included patients from specialized clinics with a high prevalence of severe fibrosis, the applicability of combined strategies must be tested in different settings. [154].

The optimal management of the increasing population of NAFLD in T2DM depends on initial screening in primary care, endocrinology, and diabetology settings, followed by staging of fibrosis (Figure 1).

B-mode ultrasonography and blood biomarkers are cost-effective methods for initial evaluation. FIB-4 has proven reliable in diabetic patients and has been included in current guidelines as a tool to predict liver fibrosis in primary care. Transient elastography is increasingly used in gastroenterology and hepatology settings for both liver steatosis and fibrosis assessment. According to published data, CAP can be used for screening fatty liver in T2DM, although it has shown limited performance for steatosis quantification. Further research is required in populations with obesity and severe steatosis, where the limited performance of CAP was reported. VCTE has proven to be a valuable non-invasive method for the detection of advanced fibrosis. Combining TE with blood-based markers could potentially increase diagnostic accuracy, although data from small sample size studies need to be replicated in larger populations. The potential of TE for monitoring NAFLD progression or treatment response deserves further research. Published data have demonstrated the superiority of MR imaging modalities over ultrasound methods in discriminating between steatosis and fibrosis stages, quantifying small variations in liver fat content, and selecting high-risk patients who will benefit from therapeutic interventions. However, studies addressing the cost-effectiveness of different imaging methods for NAFLD monitoring in diabetic patients are lacking.

Newly developed tests, combining various serum markers and imaging techniques, have been recently proposed to increase accuracy in detecting high-risk patients with progressive NASH. The NIS4 (including miR-34a-5p, alpha-2 macroglobulin, YKL-40, and glycated hemoglobin) and the MACK-3 (including AST, glucose, insulin, and CK 18) are blood-based diagnostic tools that have proven useful in detecting “fibrotic NASH” (defined as NASH + NAS ≥ 4 + fibrosis stage ≥ 2) [155,156]. The LIVERFASt is an artificial intelligence-based algorithm that combines age, gender, weight and height with ten biomarkers (AST, ALT, GGT, total bilirubin, total cholesterol, triglycerides, fasting glucose, alpha2-macroglobulin, haptoglobin, and apolipoprotein A1) for the assessment of liver fibrosis, steatosis, and inflammatory activity. It was developed as a non-invasive tool for the diagnosis and monitoring of NALFLD, NASH, hepatitis B, and hepatitis C. Its performance in estimating severe fibrosis is similar to TE and better than FIB-4. The applicability of the algorithm could be extended to primary care as a screening tool for NAFLD/NASH in populations at risk, including patients with T2DM [157].

The FibroScan-AST (FAST) score combines LSM and CAP measured by FibroScan with AST [158], while the MAST score combines MR imaging modalities (MRI-PDFF and MRE) with AST. The MAST score outperformed other scores (NFS, FIB-4, and FAST) in identifying patients with NASH and significant fibrosis [159]. The cTAG score represents another combination of imaging (cT1) and blood biomarkers (AST and fasting glucose) that demonstrated increased accuracy in screening patients with progressive NASH (AUROC of 0.90) [160]. Combined strategies for the diagnosis of progressive NASH need to be tested in larger trials before they can be recommended for routine use in clinical practice.

## 5. Conclusions

Recognizing the potential and limitations of non-invasive methods, the best diagnostic strategy in diabetes-associated NAFLD depends on the patient’s particularities (e.g., obesity, metallic implants), available technology, associated costs, local expertise, and the purpose: screening in primary care practice, risk stratification, monitoring after lifestyle and therapeutic interventions, or research trials. The sequential approach and rational use of blood-based and imaging biomarkers that have already proven their practical applicability in a clinical setting, as well as the appropriate selection of patients who need follow-up and therapy, will improve NAFLD outcomes.

Although numerous reports have demonstrated the utility of biomarkers for evaluating treatment response in NAFLD, the correlation between dynamic changes of imaging parameters, improvement in liver histology, and the clinical course of the disease needs to be investigated in longitudinal studies. Optimal thresholds should be defined before biomarkers can be used as surrogates for histologic improvements in therapeutic trials. Advanced imaging methods and combined scores developed to assess liver inflammation and progressive NASH, as well as their potential role in guiding treatment, are under evaluation. Many of them are more suitable in a research setting and require validation in large cohorts. Additionally, the cost-effectiveness of various non-invasive methods deserves further research, which may assist practitioners in choosing the best strategy for NAFLD diagnosis and monitoring.

## Figures and Tables

**Figure 1 life-13-02262-f001:**
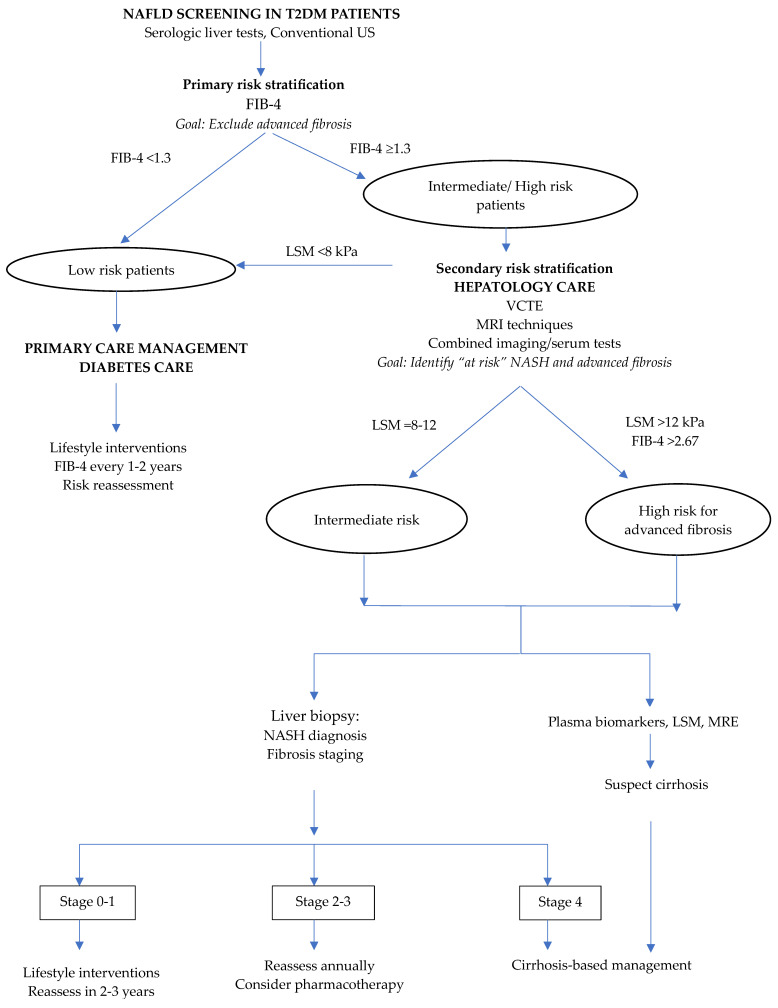
Strategy for NAFLD evaluation in T2DM patients. US: ultrasonography; VCTE: vibration-controlled transient elastography; LSM: liver stiffness measurement; MRI: magnetic resonance imaging; MRE: magnetic resonance elastography; NASH: non-alcoholic steatohepatitis.

**Table 1 life-13-02262-t001:** Usefulness of imaging techniques for non-invasive assessment of NAFLD in diabetic patients.

Imaging Techniques	Strengths	Limitations	Applicability
**Conventional B-mode US**	Lower cost Increased availability High performance for moderate/severe steatosis	Operator and machine dependent Reduced performance for mild steatosis Cannot assess inflammation (NASH) and early fibrosis Interobserver and intraobserver variability Difficult evaluation in obese patients	Screening of NAFLD in clinical settings Assessment of portal hypertension, specific features of cirrhosis
**Semi-quantitative US methods** **(Hepatorenal index, Hamaguchi score)**	Computer- assisted: eliminate operator subjectivity Promising results for liver steatosis assessment	Reduced performance in concomitant fibrosis	Data from research trials Reduced availability in current practice
**Quantitative US methods** **(VCTE)**	Point-of-care technique Lower cost compared to other imaging methods Accessibility, reproducibility High accuracy for the detection of advanced fibrosis	Not feasible in the presence of ascites Reduced performance for moderate/severe steatosis Limited performance in grading steatosis in NAFLD Limited performance to differentiate mild/moderate fibrosis Hepatic inflammation, obstructive cholestasis, congestive heart failure, postprandial status, exercise increase LSM CAP confounding factors: diabetes, obesity	High applicability in clinical practice Screening tool Risk stratification- fibrosis severity Patient monitoring
**CT**	Increasing availability Good performance for moderate to severe steatosis, pre-cirrhotic fibrosis and cirrhosis	Limited accuracy in detecting low-grade steatosis Pitfalls: iron overload, amiodarone, iodine contrast, glycogen overload, hepatitis Risk of ionizing radiation exposure	Unsuitable for screening and patient monitoring in routine clinical practice
**MRI techniques**	High accuracy and reproducibility Volumetric assessment of liver fat content (MRI-PDFF) Superiority in fibrosis quantification over US-based methods (MRE) Promising results in diagnosing high-risk NASH	Significant cost Limited availability Complexity of imaging acquisition and data processing (time and expertise requirement) Patient-related = claustrophobia, implanted devices	Increasing clinical and research applicability Risk stratification Patient monitoring Assessment of treatment response in research trials

US: ultrasonography; NASH: non-alcoholic steatohepatitis; NAFLD: non-alcoholic liver disease; VCTE: vibration-controlled transient elastography; CT: computed tomography; MRI: magnetic resonance imaging; MRI-PDFF: magnetic resonance imaging-proton density fat fraction; MRE: magnetic resonance elastography.

**Table 2 life-13-02262-t002:** Performance of imaging methods for the assessment of liver changes in NAFLD.

Method	Authors/Reference	Reference Method	Meta-Analysis	Performance
**Conventional US**	Saadeh et al. [66]	LB	No	100% Se for moderate-severe steatosis (>33% liver fat content) Unable to distinguish NASH
**Conventional US**	van Werven et al. [67]	LB	No	65% Se, 77% Sp for liver steatosis
**Conventional US**	Hernaez et al. [68]	LB	No	84.8% Se, 93.6% Sp for moderate to severe steatosis
**Conventional US**	Bohte et al. [69]	LB	Yes	73.3%–90.5% Se, 69.6%–85.2% Sp for steatosis Limited performance in mild steatosis
**Hepatorenal index (HRI)**	Shiralkar et al. [70]	LB	No	HRI ≥ 1.34: 92% Se, 85% Sp, 94% NPV, 79% PPV for liver fat > 5%
**Hamaguchi US score**	Ibacahe et al. [71]	MRS	No	Score ≥ 4: 78% Se, 85% Sp, AUROC 86% for steatosis
**CAP**	Karlas et al. [72]	LB	Yes	AUROC 0.823 for S ≥ 1 (cut-off 248 dB/m) AUROC 0.865 for S ≥ 2 (cut-off 268 dB/m) AUROC 0.882 for S = 3 (cut-off 280 dB/m)
**CAP**	Eddowes et al. [73]	LB	No	AUROC 0.87 for S ≥ 1 (cut-off 302 dB/m) AUROC 0.77 for S ≥ 2 (cut-off 331 dB/m) AUROC 0.70 for S = 3 (cut-off 337 dB/m)
**CAP**	Pu et al. [74]	LB	Yes	87% Se, 91% Sp, AUROC 0.96 for S ≥ 1 85% Se, 74% Sp, AUROC 0.82 for S ≥ 2 76% Se, 58% Sp, AUROC 0.70 for S = 3
**CAP**	Petroff et al. [75]	LB	Yes	AUROC 0.819 for S ≥ 1 AUROC 0.736 for S ≥ 2 AUROC 0.711 for S = 3
**CAP**	Caussy et al. [76]	MRI-PDFF	No	AUROC 0.80 for liver fat ≥ 5% on MRI (cut-off 288 dB/m) AUROC 0.87 for liver fat ≥ 10% on MRI (cut-off 306 dB/m)
**US-LSM**	Kumar et al. [77]	LB	No	AUROC 0.82 for F ≥ 1 (cut-off 6.1 kPa) AUROC 0.85 for F ≥ 2 (cut-off 7.0 kPa) AUROC 0.94 for F ≥ 3 (cut-off 9.0 kPa) AUROC 0.96 for F = 4 (cut-off 11.8 kPa) 95% NPV to rule out advanced fibrosis
**US-LSM**	Eddowes et al. [73]	LB	No	AUROC 0.77 for F ≥ F2 (cut-off 8.2 kPa) AUROC 0.80 for F ≥ F3 (cut-off 9.7 kPa) AUROC 0.89 for F = F4 (cut-off 13.6 kPa)
**US-LSM**	Wong et al. [78]	LB	No	91% Se, 75% Sp, 52% PPV, 97% NPV for F ≥ F3 (cut-off 7.9 kPa)
**US-LSM**	Kwok et al. [79]	LB	Yes	79% Se, 75% Sp for F2 85% Se, 82% Sp for F3 92% Se, 92% Sp for F4
**CT**	Bohte et al. [69]	LB	Yes	57% Se, 88% Sp for mild steatosis 72% Se, 94.6% Sp for moderate to severe steatosis
**CT**	Huber et al. [80]	LB	No	83% Se, 76% Sp for pre-cirrhotic fibrosis 88% Se, 82% Sp for cirrhosis
**MRI-PDFF**	Gu et al. [81]	LB	Yes	AUROC 0.98, pooled Se 0.93, pooled Sp 0.90 for S ≥ 1, AUROC 0.91, pooled Se 0.94, pooled Sp 0.74 for S ≥ 2 AUROC 0.90, pooled Se 0.74, pooled Sp 0.87 for S = 3
**MRI-PDFF**	Middleton et al. [82]	LB	No	83% Se, 90% Sp for S ≥ 2 (PDFF thresholds 16.3%) 84% Se, 90% Sp for S = 3 (PDFF thresholds 21.7%)
**MRI-PDFF vs CAP**	Park et al. [83]	LB	No	AUROC 0.90 vs. 0.70 for S ≥ 2 AUROC 0.92 vs. 0.73 for S = 3
**MRE vs TE**	Park et al. [83]	LB	No	AUROC 0.89 vs. 0.86 for F ≥ 2 AUROC 0.87 vs. 0.80 for F ≥ 3 AUROC 0.87 vs. 0.69 for F = 4
**MRE vs TE**	Hsu et al. [84]	LB	Yes	AUROC 0.87 vs. 0.82 for F ≥ 1 AUROC 0.92 vs. 0.87 for F ≥ 2 AUROC 0.93 vs. 0.84 for F ≥ 3 AUROC 0.94 vs. 0.84 for F = 4 (MRE thresholds: 2.61, 2.97, 3.62, and 4.69 kPa)

LB: liver biopsy; Se: sensitivity; Sp: specificity; NPV: negative predictive value; PPV: positive predictive value; AUROC: area under the receiver operating characteristic curve; S: steatosis; F: fibrosis; CAP: controlled attenuation parameter; US-LSM: ultrasound liver stiffness measurement; TE: transient elastography; CT: computed tomography; MRI: magnetic resonance imaging; MRS: magnetic resonance spectroscopy; MRI-PDFF: magnetic resonance imaging-proton density fat fraction; MRE: magnetic resonance elastography.

## Data Availability

No new data were created or analyzed in this study. Data sharing is not applicable to this article.

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
