# Peer review of "Non-Invasive Diagnostic of NAFLD in Type 2 Diabetes Mellitus and Risk Stratification: Strengths and Limitations"

_life, 2023, doi:10.3390/life13122262_

Round 1

Reviewer 1 Report

Comments and Suggestions for Authors

In this review Boeriu A et al summarize recent available data about current options for non-invasive assessment of NAFLD in patients with type 2 diabetes. I appreciate the effort of the authors to provide a complete picture in a field that is in continue up-dating. Although the Journal does not provide any length restriction for submitted manuscripts in my opinion this is too long to follow and the authors somhow lost the focus on type 2 diabetes. I suggest revising the text to obtain a concise manuscript possible focusing on the role of combination strategy for diagnosis and management of NAFLD in type 2 diabetes. Moreover, Figure 1 which reports the suggested algorithm for diagnosis and follow-up of patients with T2D, is not clear and need to be completely modified.

Reviewer 2 Report

Comments and Suggestions for Authors

This review article provides a comprehensive overview of the latest research and guidelines on diagnosing and managing non-alcoholic fatty liver disease (NAFLD) in patients with type 2 diabetes mellitus (T2DM). It discusses the accuracy and clinical utility of various noninvasive blood tests and imaging modalities like ultrasound, MRI, and CT for evaluating liver steatosis, inflammation, and fibrosis, using liver biopsy findings as a reference standard. The paper highlights the strengths and limitations of these methods for screening, risk stratification, treatment monitoring, and their appropriate applications in clinical practice versus research trials for NAFLD assessment in diabetic patients at high risk of progression. However, the review does not critically analyze potential biases in the cited studies or provide details on optimal diagnostic algorithms for NAFLD management. Furthermore, it does not compare the cost-effectiveness of emerging modalities or discuss gaps in the evidence that require future research.

However, several shortcomings need to concisely addressed:

           Potential biases in the cited studies are not critically analyzed, nor are details provided on optimal diagnostic algorithms for NAFLD management.

           The cost-effectiveness of emerging modalities is not compared, nor are gaps in the evidence that require future research discussed.

           The rarely tackled temporal variation and monitoring, e.g.,  from viremic baseline to nonviremic post-HCV cure disease stages, are not addressed.

           Some other very new diagnostic tests were not concisely pointed out in brief phrases: NIS4 score, cTAG score, MAST score, LiverFASt, FibroScan-AST (FAST) score…

Comments on the Quality of English Language

-In the introduction, “Data from epidemiologic studies show that approximately 20% of non-alcoholic fatty 314 liver disease (NAFLD) patients will develop NASH (liver inflammation and hepatocyte 325 damage), which carries the potential to progress to advanced fibrosis (20% among NASH 336 cases) and even hepatocarcinoma.”, the word “hepatocarcinoma” should be “hepatocellular carcinoma”.

 -In the section 2.1, “The Fatty Liver Index (FLI) is a simple test combining BMI, waist circumference, 1057 serum triglycerides, and GGT. Cut-off values ≥ 60 rule in fatty liver, with 86% sensitivity 1068 and 87% specificity (area under the receiver operating characteristic [ROC] curve [AUC] 1079 =0.85) [33].”, there should be a space between “=” and “0.85”.

-In the section 3.1, “The heterogeneous distribution of fat in the liver and the variability of biopsy sampling 52110 prevent accurate estimation of disease severity and adequate monitoring of disease 522 progression.”, the word “prevent” should be “prevents”.

Round 2

Reviewer 1 Report

Comments and Suggestions for Authors

I really appreciate the effort of the authors to provide a so complete revision of current available non-invasive methods for NALFD assessment in subjects with T2D.  In my opinion Figure 1 remain a bit confusing (at least lines and graphical connections would help), but overall, the paper is interesting and no further modifications are needed.